# Interaction with Nature Indoor: Psychological Impacts of Houseplants Care Behaviour on Mental Well-Being and Mindfulness in Chinese Adults

**DOI:** 10.3390/ijerph192315810

**Published:** 2022-11-28

**Authors:** Jingni Ma

**Affiliations:** School of Health in Social Science, University of Edinburgh, Edinburgh EH8 9AG, UK; jma34@ed.ac.uk

**Keywords:** restorative environment, houseplants care, well-being, mindfulness, nature relatedness

## Abstract

(1) Background: The rapid growth of urbanisation and the increased prevalence of mental problems have been concerns in China. ‘Green prescription’ such as keeping houseplants has been found to be an effective means of interacting with nature and improving mental health outcomes. The purpose of this study was to examine the psychological effects of keeping houseplants in the home and to examine how ‘connection to nature’ relates to mental well-being and mindfulness among Chinese adults living in urban areas; (2) Methods: A cross-sectional survey was completed by 421 (66.3% female) participants from 19 representative Chinese cities; (3) Results: Results revealed that participants who self-classified themselves as ‘houseplants carers’ reported higher levels of mental well-being compared with ‘non-houseplants carers’. Specifically, hours spent on taking care of houseplants, the number of houseplants, and the years of keeping houseplants were positively associated with greater levels of mental well-being and the trait of mindfulness. (4) Conclusions: The findings of the current study have implications for use of therapeutic horticulture for people who are seeking to improve their mental well-being and mindfulness in urban China.

## 1. Introduction

Keeping houseplants indoors appears to generate psychological well-being benefits [1]. Houseplant care in this study refers to the behaviour of keeping and growing pot plants in one’s place(s) of residence. The plants (e.g., flowers, vegetables, green plants, succulent plants) in question may have been cultivated from an early stage (e.g., as seedlings), or they may have been raised entirely from seed. Individuals who regularly take care of their plants over six months are recognised as ‘houseplants carers’ within the current study. Houseplant care is a way of connecting oneself to nature especially when staying indoors.

### 1.1. Houseplant Care, Mental Well-Being, and Mindfulness

Houseplants are part of our natural environment. The impact of keeping houseplants on being connected with nature and on mental health has been the object of increasing academic attention in recent years. It has been demonstrated that there is a positive relationship between the accessibility of green spaces, well-being, and mental health [2,3,4]. Access to nature improves human wellbeing by mitigating attention fatigue and reducing stress. There is a reciprocal connection between mindfulness and connectedness to nature [5], and natural environments can strengthen the effects of mindfulness-based interventions [6].

The potential mechanism here may be that natural environments encourage one’s meditative awareness and strengthen the symbolic relationships between nature and human beings. The more time one spends in natural surroundings, the more one will feel an affinity with the natural environment (e.g., mountains, water, clouds, plants) which in turn fosters one’s trait mindfulness [7]. This process can also be explained by the biophilia hypothesis [8], which suggests that humans have an inherent need to be connected with nature, which in turn reflects our natural affinity for other living creatures.

China is now experiencing a rapid process of urbanisation with transformative sociocultural implications. It is indicated that 300 to 400 million Chinese will become urban inhabitants by 2050, bringing the country’s urbanisation rate to around 75% [9]. In addition, the prevalence of ‘mental disorders’ in China was 9.3% in 2019 [10]. This can also be a low-cost and self-guided approach to alleviating certain psychological symptoms as a substantial number of Chinese people live in apartments without access to gardens. For example, one experimental study found that even a short duration of exposure to flora (five minutes in the presence of plants) can effectively reduce one’s level of stress as shown in the Chinese sample [11]. Furthermore, greater number of indoor plants provides more opportunities for people to acquire visual exposure to flora, which reduce people’s stress level [12,13].

It should be noted that simply spending time with plants is not the same as active planting behaviour. It may be hard to detect whether the psychological benefits that participants gained from interaction with plants were related to their passive rather than active interaction with plants. For instance, some experimental laboratory studies, have identified psychological benefits of simply viewing plants [14,15]. In addition, previous studies of the effect of indoor planting on psychological well-being have been largely based on the workplace settings rather than in people’s homes. For example, past research demonstrated that employees’ stress is mitigated by higher quantities of indoor flora in indoor working environments [16,17]. Such studies, however, have predominantly focused on the viewing of indoor plants or being in the same environment as plants in a passive manner, rather than involving in consistent behaviour on growing plants [14,18]. Conversely, it remains unclear whether it is really feasible to promote houseplant care behaviour on an everyday basis. It is still unknown whether the levels of engagement in keeping houseplants (e.g., frequency and year of house planting, stages of plant ownership) impact on Chinese urban residents’ mental well-being and mindfulness levels, and any reliable information about the preferable characteristics of houseplants in urban China can be provided.

### 1.2. The Current Research

The current study sought to understand the differences of mental well-being and mindfulness amongst houseplants carers and non-houseplants carers, and test the relationship among houseplants care engagements, degree of nature relatedness, mental well-being, and trait mindfulness. The research questions were as follows:How do mindfulness levels and mental well-being differ amongst people at different stages of houseplant care behaviours?How do engagements of houseplant care behaviours (The actual houseplants care behaviour in this study has three dimensions, namely: the hours spent taking care of plants in one week (behaviour 1), the total numbers of plants (behaviour 2) taking care of, and the years spent planting indoors (behaviour 3)) relate to mental well-being and mindfulness levels?How does the degree of nature relatedness relate to mental well-being and mindfulness levels?

The hypotheses were that (a) participants who recognise themselves as houseplants carers will report greater levels of mental well-being and mindfulness than those who classified themselves as non-houseplants carers; (b) more hours spent taking care of plants would be associated both with higher levels of mental well-being and with greater mindfulness; (c) greater numbers of plants kept indoors are associated with higher levels of mental well-being and mindfulness; (d) longer years of conducting houseplants care are related to greater levels of mental well-being and mindfulness; (e) greater levels of nature relatedness would be associated with greater mental well-being and mindfulness.

## 2. Materials and Methods

### 2.1. Design

This is an online questionnaire-based cross-sectional study. Questionnaires are an effective and practical tool for harvesting information and are widely used in psychological research.

### 2.2. Participants

According to power calculations using G*Power software [19], this study required a minimum of 400 participants to ensure the statistical power of conducting Pearson’s correlation test and MANOVA test with medium effect size (*f*^2^ = 0.25). In total, 430 participants initially participated in this study. Nine survey responses were excluded due to large amounts of missing data (up to 90%). This study achieved a sample comprised of 421 participants and most of them aged between 26 to 34 years (66.3% female).

### 2.3. Inclusion and Exclusion Criteria

Participants had to be: (1) Chinese residents; (2) fluent in reading written Chinese; (3) aged 18 or over; (4) living in an urban area of the targeted Chinese cities. We include participants who only take care of indoor houseplants. Participants should not have been involved in community or outdoor gardening on a regular basis because they may have higher levels of nature relatedness and better mental health than our target participants.

This study used a stratified sampling design to recruit representative participants. An established sampling model was adopted and modified for this study, as specified by the Chinese General Social Survey (CGSS) [20]. China contains twenty-six provincial capitals and four municipalities, and these provided the primary sampling units (PSU). Therefore, two sampling frames were used due to inconsistent development in the economy and geography of Chinese cities. The first sampling frame contains four municipalities, whereas the second sampling frame contained the two provincial capitals with three clusters (western, central, and eastern regions). Following the stratified sampling design set out by the CGSS, nineteen cities were targeted which were representative of the general profile of Chinese cities in terms of economy, demographics, and geography. They were selected to reflect both the developing cities in the three main regions of China, and the larger, more developed cities. The residents of these cities may reasonably be seen as representative of urban citizens in mainland China (see Appendix A).

There were five categories for different occupations, namely: student, full-time job, part-time-job, unemployed and retired. Meanwhile, the selected cities were divided into four groups: municipalities, western region, central region, and eastern region. Ideally, at least 20 participants (for a minimum sample size of 400) would be recruited from each layer (i.e., 4 × 5 = 20).

### 2.4. Measures and Procedure

The Jisc Online Survey was utilised and the survey was distributed to the targeted nineteen Chinese cities between the 30 May and 30 June 2020 and was promoted through social media (e.g., Weibo; Online forums about indoor planting in Baidu; WeChat). The questionnaire consisted of seven parts as follows:

Demographic information: Questions regarding marriage and employment status, age, educational background, gender, and city of residence were asked.

Profile of Plant Ownership and Houseplant Care Behaviour: Descriptive questions about participants’ houseplant care behaviour were included. Firstly, questions were asked to allocate participants into different stages of houseplant behaviour based on the Transtheoretical Model (or Stages of Change [21]). The TTM is an integrative model of behavioural change which can be employed to determine participants’ current house planting behaviour in different stages of change, namely: precontemplation, contemplation, preparation, action and maintenance [21].

Participants chose a category which most appropriately described their behaviour (see Appendix A). If participants self-recognised that they have already been a ‘house planter’ (in the stage of action and maintenance), they were also asked to report weekly hours spending on taking care of plants, numbers of plants taking care of at home, and years of keeping houseplants. Otherwise, participants who were not house planters could report a number 0 in each of this question or leave it as blank.

Additionally, to better understand the ownership of houseplants of the participants, preferable characteristics of plants were asked (see Appendix A). This measure comprised six items, which were rated on a five-point Likert scale from “extremely unimportant” to “extremely important”. Items in this measure included: “nice scent of plant”, “small size of plants”, and “green coloured plants”, and which were items adopted from a previous study [17].

Nature Relatedness: The Nature Relatedness Scale (NR-6) was used to examine participants’ degree of cognitive and affective connection with nature. The higher scores of relatedness with nature indicated stronger subjective feelings of nature connection [22]. The scale demonstrated a robust internal consistency and contained six short questions regarding one’s cognition, affect and experiences of nature connection. The NR-6 scale has proven robust in terms of both internal consistency and reliability (α = 0.83).

Mental Well-being: The Chinese version of the Short Warwick-Edinburgh Mental Well-being Scale (S-WEMWBS) was adopted. The WEMWBS has shown itself to be a valid and reliable measurement of mental health and well-being. Likewise, testing of the Chinese version of the short WEMWBS has led to its validation and the confirmation of its reliability (α = 0.89) [23].

Mindfulness: The Five Facets Mindfulness Questionnaire (FFMQ) was used to measure trait mindfulness levels. The measure consists of 39 items and was developed from a variety of mindfulness-based measurements [24]. The FFMQ can explore an individual’s mindfulness in five areas, namely: observing, describing, acting with awareness, ‘non-judgemental to inner experience’, and ‘non-inactivity to inner experience’ [24]. The Chinese version of the FFMQ has demonstrated reliable psychometric properties for all subscales except for the non-reacting (Cronbach’s α: Observing = 0.75, Describing = 0.84, Actaware = 0.79, non-judging = 0.66, non-reacting = 0.45) [25].

### 2.5. Data Cleaning and Analysis

SPSS 24 was utilised for the purpose of statistical analyses. As each question of the survey required an answer, no missing data resulted. When statistical analysis was performed, a few missing data were identified, which was replaced in version 24 SPSS software. Initially, descriptive statistics, including percentage and mean, provided an overview of each stage of change in relation to age, gender, occupation, city of residence. The Shapiro–Wilk test was also used to check the normality of all the continuous variables.

As not all the continuous variables in the current study were normally distributed (*p* < 0.05), for the first and the second research questions, Spearman’s correlation was conducted to examine the relationship among scores on houseplants care behaviour, nature relatedness, mental well-being, and mindfulness levels. The stage of planting behaviour (within five possible categories) provided the independent variable for the final research question, the dependent variables were mental well-being and mindfulness levels. Multivariate analysis (MANOVA) was carried out to assess disparities in levels of mindfulness and well-being among five ‘stage of planting’ groups of participants.

### 2.6. Ethical Consideration

Participants voluntarily took part in the study and their data was anonymised. Informed consent was obtained from all individual adult participants included in the study. All procedures performed in this study involving human participants were in accordance with the 1964 Helsinki Declaration and comparable ethical standards and its later amendments or comparable ethical standards. This study gained ethical approval by the Clinical and Health Psychology Ethics Committee of the University of Edinburgh on 29 May 2020 (Reference number: CLIN776).

## 3. Results

### 3.1. Demographics and Exploratory Statistics

Darker colours on the map (Figure 1) represent greater numbers of participants. This study recruited 421 participants from the targeted cities in mainland China. The average number of participants per city was 21. There were variations, however, with the fewest individuals supplied by Jinan (*n* = 11), and a relatively large sample from Guangzhou (*n* = 57). In terms of ideal sampling, the objective was for each city to provide 20 participants. Discrepancies in participant numbers in certain cities may have been due to the nature of the online survey; obtaining an exactly equal number of individuals for each locale was always unrealistic. The researcher conducted the same recruitment method in each city (e.g., finding potential candidates, distributing the survey on social media in each city). The rate of response was stronger in southern and eastern coastal areas than in northern or inland regions.

Demographic information regarding the participants is provided in Table 1. The survey participants included a high proportion of female, well-educated and relatively young individuals. According to an ANOVA test, there was no significant difference in mental well-being between residents of the 19 Chinese cities: F (18, 386) = 1.33, *p* = 0.17. Similarly, there were no statistically significant differences in mindfulness levels between residents of the Chinese cities; F (18, 402) = 0.51, *p* = 0.96.

### 3.2. Stages of Houseplant Care Behaviour and Plant Ownership

Overall, 43% of individuals who recognised themselves as non-planters (in the stages of pre-contemplation, contemplation, and preparation stages) elected to participate, which was surprising. By contrast, over half (57%) of participants self-reported themselves as house planters based on the description of Stages of Change Model. Among the planters, they spent 1.28 h to take care of their houseplants on average (SD = 2.90); they owned 5.12 numbers of houseplants on average (SD = 8.62); and these houseplant-owners have already kept their plants for 4.17 years on average (SD = 6.31). Table 2 also shows mean and SD of dependent variable on five stages of changes in houseplants care.

Figure 2 indicates the preferable characteristics of houseplants for participants to take care of, 44.42% of participants believed nice-scented plants were important and very important to them; conversely, 40.62% of participants believed that this was unimportant to them. For green-coloured plants, 46.08% and 40.62% of the participants believed it was important, or unimportant, respectively, for them to consider keeping. Additionally, more participants (39.19%) believed the small size of plants was unimportant to them than those who believed it was important (34%). Similarly, more than half (57.72%) of participants believed the texture of the plants was unimportant to them. Notably, participants concerned about how good the plants looked (81%), the ease of growing (84.80%) and cheaper cost of purchasing and maintaining their plants (54.16%), were important to very important to them. In conclusion, when keeping houseplants, Chinese adults preferred to take care of plants which were good-looking, easy to grow, and cheap to purchase.

### 3.3. How Do Mindfulness Levels and Mental Well-Being Differ amongst People at Different Stages of Houseplant Care Behaviour?

A MANOVA was used to test group differences between house plant behaviour groups (between subject variable) and mental well-being and mindfulness level (see Table 3).

The results showed, Pillai’s trace = 2.04, F (24, 1592) = 2.04, *p* < 0.01. η^2^ = 0.03, implying that 3% of the variance in the dependent variable was accounted for by stage of change. For mental well-being, significant houseplant group differences were found F (4, 400) = 6.96, *p* < 0.001, η^2^ = 0.07. Post hoc test showed that mental wellbeing scores of the pre-contemplation group increased and was statistically different from action (*p* < 0.01) and maintenance (*p* < 0.01) groups. The contemplation group was significantly different and increased from action (*p* = 0.05) to maintenance (*p* = 0.04) groups on mental wellbeing as well. Mental wellbeing scores of the preparation group were not significantly different from any other houseplant groups.

Furthermore, as Figure 3 below shows, participants who identified themselves in latter stages of houseplants care behaviour evinced a greater level of mental well-being. From the stage of pre-contemplation through to the action stage of home planting, levels of well-being rose sharply, and this increase persisted in a stable manner from the stage of action to that of maintenance.

There was a significant houseplant group difference for overall trait mindfulness, F (4, 400) = 3.85, *p* < 0.01, η^2^ = 0.04. The scores increased from the stage of pre-contemplation to that of preparation (determination), with a peak value in the preparation stage. Subsequently, the mindfulness levels increased, and remained stable in the contemplation and maintenance stages (see Figure 4). A post hoc test indicated that the overall mindfulness scores in the pre-contemplation group were significantly different from those in contemplation (*p* = 0.01), preparation (*p* < 0.01), action (*p* < 0.01) and maintenance (*p* < 0.01) groups. Moreover, the contemplation group did not show significant differences to any other groups on the score of overall mindfulness.

For each subscale of FFMQ, results shown that only for observing (F = 3.61, *p* = 0.01) and for describing (F = 2.45, *p* = 0.05) were there statistically significant differences between the five houseplant behaviour groups (see Figure 5).

Post hoc tests indicated that levels of observing in the pre-contemplation group were significantly different; it decreased from pre-contemplation to preparation, and then increased to action (*p* = 0.04) and maintenance (*p* < 0.01) groups; contemplation and preparation groups were not significantly distinct from other groups on observing scores. Additionally, for describing levels, similarly, the pre-contemplation group was only significantly different from action (*p* = 0.04) and maintenance (*p* = 0.02) groups.

In summary, the results show that scores of mental well-being, overall mindfulness, and its subscales of observing and describing scores, were highest in the preparation to action stages of house planting behaviour, and the group differences between pre-contemplation and action, as well as maintenance groups, were significant on measured outcomes. It suggests that there was a significant difference on levels of mental well-being and mindfulness between non-house planters and house planters.

### 3.4. Engagement of Houseplant, Mental Well-Being and Mindfulness

Table 4 below shows the results of analyses to test associations between levels of engagement of caring for houseplants, degrees of mental well-being, nature relatedness and trait mindfulness. The results indicated that having spent more hours in the last week, owning greater numbers of houseplants, and spending longer years of keeping houseplants, are statistically significant and correlated with greater levels of nature relatedness, mental wellbeing, and trait mindfulness. Furthermore, higher levels of nature relatedness were significantly associated with greater degrees of mental well-being and mindfulness. It is notable that the effect magnitude of these associations was sizeable (i.e., r = 0.31; 0.34; see more details in Table 4).

## 4. Discussion

The present study investigates the psychological impact of indoor houseplant care behaviour on mental well-being, trait mindfulness, and nature relatedness amongst Chinese adults living in urban areas. This study involved 421 samples (66.3% female), and most of them were young and well-educated. Over half of the participants categorised themselves as houseplants carers, and they reported that on average, they spent 1.28 h on taking care of their houseplants in the last week, owned 5.12 pots of plants, and had spent 4.17 years keeping houseplants.

Firstly, this study indicates that mental wellbeing and mindfulness levels were significantly different among participants at differing stages of houseplants care behaviour. Mental well-being was higher among houseplants carers than non-houseplants carers, participants who are currently growing houseplants (in Action group) reported the highest levels of well-being in comparison with other four behavioural groups of participants For the mindfulness level, participants who prepared for growing houseplants reported the highest levels of overall mindfulness among other behavioural groups; nonetheless, participants who are actively growing houseplants (in Action and Maintenance groups) reported higher levels of overall mindfulness than non-houseplants carers (in Precontemplation and Contemplation groups). In other words, the participants were at their most mindful when they were preparing to begin keeping houseplants. That might be because only smaller numbers were involved at some stages, which would likely affect the reliability of mean scores for those stages.

Secondly, participants in the contemplative stage of keeping house plants were more likely to look around them, rather than attempting to act precipitously. Moreover, people preparing to take care of houseplants showed the highest levels of ‘non-judging’, ‘describing’, and ‘acting with awareness’. Again, the data reflected a connection between the stages prior to starting houseplants care, and enhanced mindfulness. The possible reasons for this are as follows: (a) the boundaries between houseplants carers than non-houseplants carers were too detailed. There were five ‘types’ of houseplants carers in this study, which may have created ambiguity regarding the differences between planters and non-planters in terms of mindfulness levels. Moreover, (b) the measurement of mindfulness the study deployed is commonly used to test mindfulness changes, before and after a mindfulness-based intervention. Further research is needed to examine differences in mindfulness before and after engaging in plant-cultivation behaviours/interventions. In any case, to investigate the effect of domestic plant cultivation on rumination, meditation and mindfulness in greater detail, interviews will be required.

Thirdly, for the houseplant carers in urban China, the findings suggest the more hours spent taking care of plants in the last week, keeping greater numbers of houseplants, spending more years taking care of plants, and possessing greater levels of nature connectedness were positively correlated with greater levels of mental well-being and trait mindfulness. These findings coincide with previous studies of the relationship between horticultural activities and mental well-being [26]. Further studies can test the role of mindfulness in the relationship between houseplants care behaviour and improvement of mental wellbeing by using an experimental research design, to reveal the mechanism of improvement of mental well-being via mindfulness.

The findings are novel because these add to the body of literature on the benefits of houseplants for mental health and mindfulness. Cultivating one’s own mindfulness via the literal ‘cultivation’ of plants is a reasonable and intuitive concept that may attract many individuals. For instance, one project (‘Green Mindfulness’) at a Canadian university integrated plants with mindfulness practice, aiming to improve students’ well-being [27]. However, the relationship between improved mindfulness and indoor house planting has been sparsely researched.

Lastly, this study provides an insight on the preferred characteristics of plants amongst Chinese adults. Ease of growing is the most important factor that people, especially beginners, consider when raising plants at home. The aesthetics of the plants is also an important factor that people consider. Overall, less than half of the participants believed that green coloured, small sized, and soft textured plants were important for them to plant. This study suggests that ease of growing, aesthetics, and affordability are the important aspects Chinese adults consider when keeping houseplants. This finding can be incorporated with further self-guided planting interventions for adults in China. Further relevant studies are encouraged to collect data regarding the preferable plant species that urban residents like to grow indoor, which may also provide valuable knowledge for designing and promoting the therapeutic use of nature indoor.

### 4.1. The Significance of the Present Study

The present study is original in its exploration of association of indoor houseplants care behaviours with Chinese city residents’ mental well-being and mindfulness level. This study concurs with earlier research regarding the well-being advantages derived from indoor plants [28]. The present study also shows that a variety of planting behaviours offer one way to enhance one’s connection with nature. The increase of self-perceived mental well-being among some participants was consistent with previous studies on the positive relationship between nature connectedness and well-being [22]. This can partially explain why participants at different stages of their planting behaviour evinced significantly different levels of well-being and mindfulness. Further studies are desirable to use experimental design to explore whether nature connection is a mediator for the improvement of mental well-being and mindfulness within home planting behaviours.

It has been demonstrated by much prior research that those horticultural activities promote human mental health in various ways. Nonetheless, few of these studies examined the relationship between planting-related behaviour and mindfulness, especially in the sense of keeping plants being deployed to cultivate mindfulness. Gardens can be excellent places to practise meditation and to become centred on the present moment [29,30]. For many Chinese urban residences, either private gardens or outdoor community gardens may not be very prevalent or common. However, the present study provides an inexpensive and easily conducted approach, which may be further developed into nature-based interventions, and even be prescribed by medical practitioners. Further research should address the therapeutic influence of plant cultivation among individuals of different nationalities and cultures.

The findings from the current study can assist with developing a rational intervention to enrich lives within the general population. Such practices and interventions may also be directed to the management of everyday stress, and not exclusively to mental or emotional pathologies.

### 4.2. Limitations

There are several limitations to the present study. Firstly, as it is a cross-sectional study, differences of mindfulness levels between planters and non-planters are still unclear. To investigate mindfulness discrepancies between these two categories, additional studies are desirable. In addition, the Stage of Change Model, which was employed in this study to theoretically identify participants’ current behaviour of house planting, contributed to unbalanced numbers of participants in each group, further relevant research can group participants into (a) actively growing houseplants; and (b) not currently growing houseplants. Secondly, although stratified sampling was used to recruit representative participants among Chinese city residents, the nature of online surveys meant that the participant numbers from each city were unequal. Third, since most participants were young and relatively well-educated, they may not be strongly representative of the wider population. In addition to the sample, the present study excludes potential participants who had regular experience in outdoor gardening, it may reduce sample size and increase the difficulties to recruit representative participants. Further studies may consider including more participants who have or have not engaged in outdoor gardening and other nature-based activities.

## 5. Conclusions

Growing houseplants is a way of connecting with nature indoor and cultivate better mental well-being and mindfulness levels in urban China. The present cross-sectional study is the first empirical study in China to confirm aspects of the houseplants care behaviours were associated with greater levels of mental health and trait mindfulness. Taking care of a greater numbers of plants and spending longer time on it were correlated with greater levels of mental well-being and mindfulness. Significant group differences between different stages of change have been found in mental well-being and trait mindfulness. The present study provides evidence that may be deployed both to illustrate the mental-health advantages of houseplants care behaviour, and to assist in the formation of mechanisms to promote this healthy behaviour in urban China.

## Figures and Tables

**Figure 1 ijerph-19-15810-f001:**
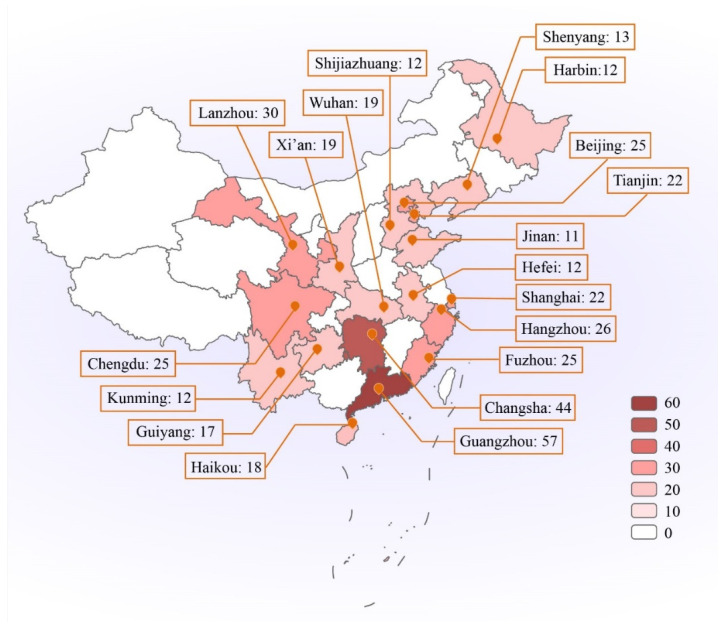
The distribution of participants in mainland China. Note. The number of participants is indicated in the map, the city of the participants’ living is marked by orange icon. The darker colour represents the greater numbers of participants.

**Figure 2 ijerph-19-15810-f002:**
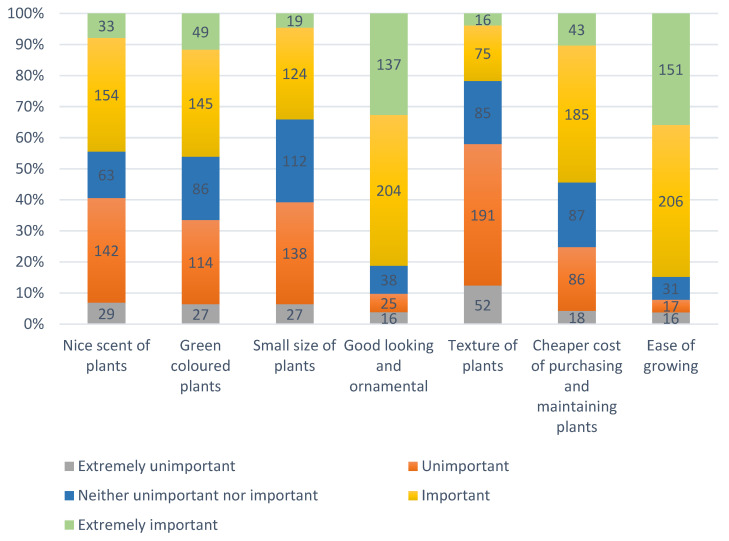
Preferable characteristics of plants.

**Figure 3 ijerph-19-15810-f003:**
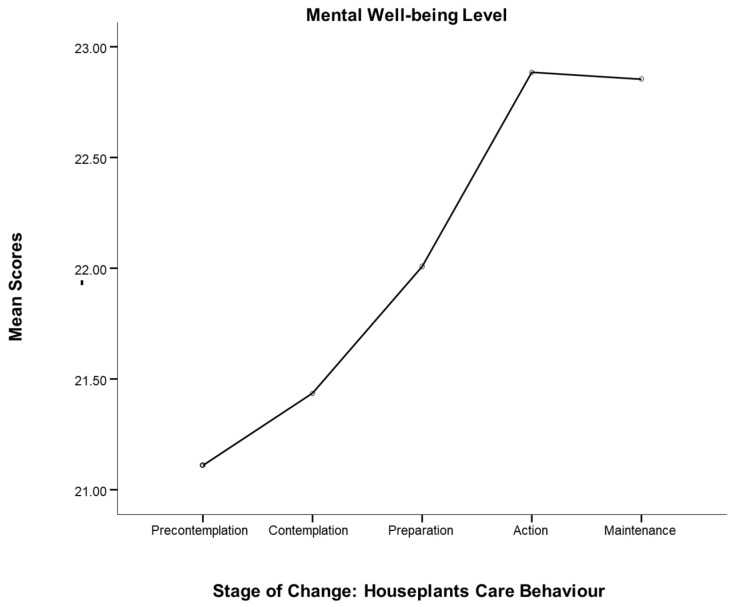
Houseplants care behaviour and mental well-being.

**Figure 4 ijerph-19-15810-f004:**
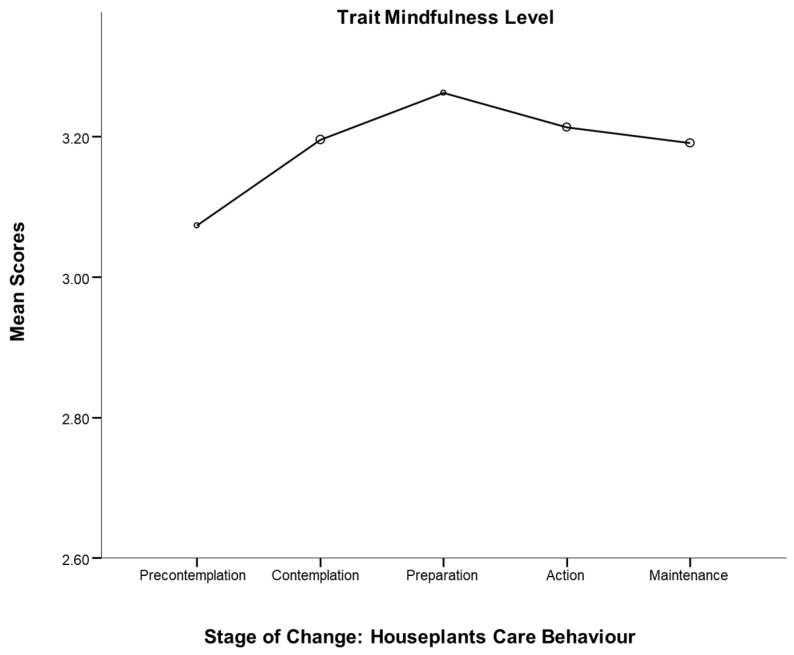
Houseplants care behaviour and mindfulness.

**Figure 5 ijerph-19-15810-f005:**
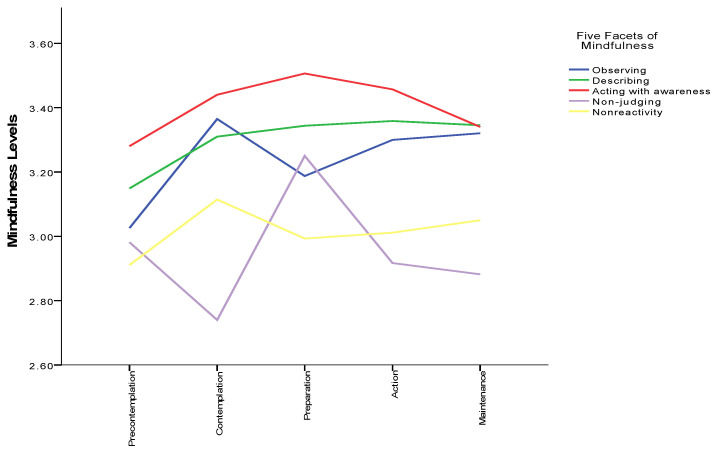
Subscales mindfulness and houseplants care behaviour.

**Table 1 ijerph-19-15810-t001:** Sample demographic information.

Classification of Variables	(*n* = 421)	
	*n*	%
Gender		
Male	133	31.6
Female	279	66.3
Prefer not to say	9	2.1
Age Range		
18–25	144	34.2
26–34	181	28.7
35–44	34	8.1
45–60	51	12.1
>60	11	2.6
Highest Level of Education		
Primary School	1	0.2
Middle School and below	10	2.4
High School and below	21	5.0
College & Bachelor’s degree	198	47.0
Postgraduate and above	191	45.4
Employment Status		
Student	115	27.3
Full-time job	247	58.3
Part-time job	6	1.4
Unemployed	21	5.0
Retired	32	7.6
Marriage Status		
Single	198	47
Dating	66	15.7
Married	147	34.9
Separated	2	0.5
Divorced or Widowed	8	1.9
Stage of Indoor Planting		
Precontemplation	136	32.3
Contemplation	25	5.9
Preparation	20	4.8
Action	75	17.8
Maintenance	165	39.2

**Table 2 ijerph-19-15810-t002:** Descriptive statistics of mental well-being and mindfulness levels on different Stage of Change groups.

Variables	Stage of Change Model on Houseplants Care	Mean	*SD*	*n*
	Total	3.80	0.76	405
Mental Well-being	Precontemplation	21.11	3.09	130
	Contemplation	21.44	2.28	24
	Preparation	22.00	3.10	19
	Action	22.89	3.24	74
	Maintenance	22.85	3.20	158
	Total	22.18	3.21	405
Overall Mindfulness Level	Precontemplation	3.08	0.29	130
	Contemplation	3.21	0.28	24
	Preparation	3.28	0.21	19
	Action	3.22	0.37	74
	Maintenance	3.19	0.33	158
	Total	3.16	0.32	405
F1_Observing	Precontemplation	3.06	0.66	130
	Contemplation	3.32	0.61	24
	Preparation	3.25	0.61	19
	Action	3.28	0.78	74
	Maintenance	3.31	0.70	158
	Total	3.22	0.70	405
F2_Desscribing	Precontemplation	3.17	0.61	130
	Contemplation	3.32	0.54	24
	Preparation	3.38	0.44	19
	Action	3.36	0.71	74
	Maintenance	3.34	0.59	158
	Total	3.29	0.61	405
F3_Acting with Awareness	Precontemplation	3.25	0.64	130
	Contemplation	3.52	0.68	24
	Preparation	3.48	0.68	19
	Action	3.49	0.70	74
	Maintenance	3.35	0.76	158
	Total	3.36	0.71	405
F4_Nonjundging	Precontemplation	2.95	0.59	130
	Contemplation	2.80	0.58	24
	Preparation	3.21	0.54	19
	Action	2.94	0.63	74
	Maintenance	2.89	0.65	158
	Total	2.93	0.62	405
F5_Nonreactivity	Precontemplation	2.94	0.61	130
	Contemplation	3.05	0.48	24
	Preparation	3.05	0.49	19
	Action	2.98	0.56	74
	Maintenance	3.03	0.53	158
	Total	2.99	0.56	405

**Table 3 ijerph-19-15810-t003:** Multivariate Tests of Differences (MANOVA) of mental well-being and trait mindfulness amongst five Stage of Change groups.

MANOVA Test						
Stage of Change on Houseplants Care	Value	*f*	Hypothesis *df*	Error *df*	*p*	Partial Eta Squared
Pillai’s Trace Wilks’ Lambda Hotelling’s Trace	0.12	2.04	24.00	1592.00	0.00	0.03
	0.88	2.06	24.00	1379.20	0.00	0.03
	0.13	2.07	24.00	1574.00	0.00	0.03

**Table 4 ijerph-19-15810-t004:** Correlations between houseplant behaviour and mindfulness, nature connectedness and wellbeing.

Variable	1	2	3	4	5	6	Mean	*SD*
NR-6		0.34 **	0.28 **	0.31 **	0.28 **	0.32 **	3.80	2.90
2. S-WEMWBS			0.41 **	0.24 **	0.21 **	0.28 **	22.18	3.21
3. CH-FFMQ				0.13 **	0.10 *	0.17 **	3.16	0.32
4. Hours of caring 5. houseplants (last week)					0.78 **	0.68 **	1.28	2.90
6. Numbers of houseplants						0.73 **	5.12	8.62
7. Years of keeping houseplants							4.17	6.31

Note. * *p* < 0.05, ** *p* < 0.01 (2-tailed). *n* = 421. Abbreviations: Nature Relatedness (NR), hours spent taking care of plants in the preceding week (behaviour 1), the total numbers of plants at home (behaviour 2) and the total years spent raising plants at home (behaviour 3).

## Data Availability

Not applicable.

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
