# Peer review of "Interaction with Nature Indoor: Psychological Impacts of Houseplants Care Behaviour on Mental Well-Being and Mindfulness in Chinese Adults"

_ijerph, 2022, doi:10.3390/ijerph192315810_

Round 1
Reviewer 1 Report
The paper is well organized and the general methodology is appropriate and accurately described. The results are clearly explained and the limits of the research are well highlighted.
If possible, based on the data of the questionnaire, I suggest including some information and details on plant species. It could be helpful information to promote a therapeutic use of nature at home.
The research assesses the impact on the mental well-being of taking care of houseplants concerning Chinese adults living in urban areas.This is an interesting topic for different sectors, such as architecture, for example concerning the design of interior spaces (schools, workplaces, residential buildings).
The article highlights the importance of taking care of plants for the well-being of users, not only considering aspects related to the visual connection with nature. The reference context is urban areas in China.The article is part of a relatively well-established research trend. As the author points out in the conclusions, the research confirms the relationship between plants and mental well-being, underlining the importance of taking care of plants for the sample of people surveyed.
In my opinion, the methodology is appropriate and described with accuracy. As an architect, I would find it interesting to have information on plant species and not only on plant characteristics (Fig. 2).The conclusions are consistent with the main research question and highlight the potential and limits of the research. The references are appropriate.
It’s necessary to check the format of the title of the paragraph, figures and table.
Author Response
Dear Reviewer,
Thank you so much for your constructive and helpful comments on my manuscripts. I have addressed your comments and uploaded a Word document, please check the attachment.
Best wishes.

Reviewer 2 Report
In addition to numerous text editing comments below, I have two primary concerns with the paper: first, on lines 116 - 117, you state, "The participants should not have been involved in community or outdoor gardening on a regular basis." But how did you determine this? Those participants who were involved in gardening activities outside the home may have had very different perspectives on engagement with nature and self-reported wellness. This is a serious shortcoming. Second, methodologically, I feel that you separated respondents into too many categories, resulting in your results being muddied (which you state in the Limitations). It would have been far cleaner for you to have two categories of a) actively growing houseplants; and b) not currently growing houseplants.
Line no.
26 “generate greater psychological well-being.” Greater than what? No reference point
31 “house planters” – I dislike the use of this term, as it sounds like some process for planting houses. Please find an alternate term.
52-53 You need an updated prediction for the percent of Chinese population expected to live in urban areas.
56-61 Bad run-on sentence; separate into two
73 Why repeat citation numbers here?
347-348 This is an example of what I’ve described above as ambiguous findings with little explanation or conjecture for their outcome.
Author Response
Dear Reviewer,
Thank you so much for your constructive and helpful comments on my manuscripts. I have addressed your comments and uploaded a Word document, please check the attachment.
Kindest regards.

Round 2
Reviewer 2 Report
I have considered all of the edits made by the author in response to my earlier review, and feel that the manuscript is now ready to be published in its present form. I greatly appreciate the author's responsiveness to my suggested changes.